# SELF-SUPERVISED VIDEO PRETRAINING YIELDS GENERAL IMAGE REPRESENTATIONS

## ABSTRACT

Videos contain far more information than still images and hold the potential for learning rich representations of the visual world. Yet pretraining on image datasets has remained the dominant paradigm for learning representations that capture spatial information. Prior attempts at video pretraining made progress towards solving video-based tasks, but did so at the cost of their image understanding capabilities. In this work we revisit self-supervised learning of image representations from the dynamic evolution of video frames. To that end, we propose a procedure for data curation that addresses the domain mismatch between video and image datasets, and develop a contrastive learning framework which handles the complex transformations present in natural videos. This simple paradigm for distilling knowledge from videos to image representations, called VITO, far outperforms all prior video pretraining methods on object detection and semantic segmentation tasks, and for the first time, closes the gap with ImageNet pretraining. Furthermore, VITO remains effective when transferring to video understanding tasks such as DAVIS segmentation and UCF-101 action recognition. Together, these results suggest that video-pretraining is now strictly more general than image-pretraining and could become the new default for learning visual representations.

## 1 INTRODUCTION

Pretraining on large image datasets has been the dominant paradigm for learning representations that understand the visual world (Krizhevsky et al., 2012; He et al., 2016). In particular, self-supervised methods which learn representations that are invariant to specific image transformations have proven very powerful, surpassing supervised pretraining on a variety of downstream tasks (He et al., 2020; Hénaff et al., 2019; Chen et al., 2020; Caron et al., 2021). Although the synthetic augmentations used in these transformations capture important image priors such as scale-, color-, and translation-invariance, they pale in comparison to the complex changes in pose and viewpoint that arise in natural videos.

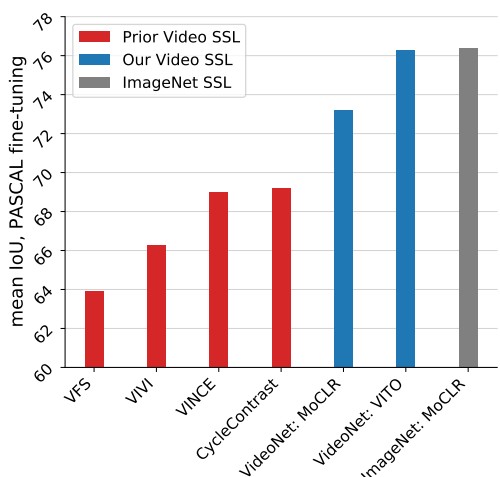

Figure 1: Closing the gap between video- and image-pretraining. Prior work learning image representations from video have lagged behind ImageNet pretraining. We propose a method for data curation, VideoNet, and a self-supervised learning objective, VITO, which together close this gap.

Therefore, one would expect that learning from videos, as opposed to images, should produce strictly more general visual representations. However, while self-supervised video representation learning has seen a variety of recent successful applications when evaluating on video-based tasks (Qian et al., 2021; Feichtenhofer et al., 2021; Toering et al., 2022; Dave et al., 2022; Feichtenhofer et al., 2022; Ni et al., 2022), it has typically done so by sacrificing performance on image classification (Gordon et al., 2020; Wu & Wang, 2021) relative to image pretraining. Furthermore, the specifics of video-representation ar-

chitectures make their comparison with image-based architectures difficult, obfuscating the role of the underlying data and learning paradigm in the quality of the resulting representations.

In this work we perform a systematic comparison of image- and video-based learning of image representations. Starting from a strong, self-supervised contrastive baseline, we find the spatial content of standard video datasets to have a detrimental effect on the quality of the resulting representations, as measured by their performance on canonical scene understanding tasks. We therefore introduce a straightforward video curation procedure—VideoNet—which aligns their class distribution with that of ImageNet, and which partially redresses the imbalance between image and video learning. Additionally, we propose three simple modifications to the standard contrastive paradigm to account for the particularities of video data: less aggressive crop augmentation, multi-scale attention pooling, and enriching view generation with natural temporal deformations. Together, these improvements yield large gains over prior video pretraining efforts on semantic segmentation on PASCAL and ADE20K and object detection on COCO and LVIS, closing the gap between image- and video-based representation learning for the first time. Notably, we maintain the expected benefits of transfer to video-based tasks (DAVIS segmentation and UCF-101 action recognition). This gives a new life to the promise of video pretraining serving as a general purpose means of learning visual representations.

## 2 RELATED WORK

**Video-based pretraining.** Many prior works have considered the problem of self-supervised representation learning for capturing spatio-temporal invariances. These span a wide range of approaches, beginning with traditional methods that leveraged temporal coherence, optical flow, and object tracking (Wiskott & Sejnowski, 2002; Hurri & Hyvärinen, 2003; Agrawal et al., 2015; Wang & Gupta, 2015; Pathak et al., 2017; Goroshin et al., 2015; Misra et al., 2016; Srivastava et al., 2015; Kulkarni et al., 2019). More recently, there have been many successful examples of approaches that leverage contrastive learning, masked autoencoding, and other self-supervised pretext tasks to learn strong video representations (Sermanet et al., 2018; Recasens et al., 2021; Qian et al., 2021; Dave et al., 2022; Dorkenwald et al., 2022; Feichtenhofer et al., 2021; 2022). However, most of these methods employ specialized video architectures and transfer to video-based tasks (action recognition, motion segmentation, object tracking, etc.) to measure the quality of the learned representations.

Natural motion-induced deformations are powerful learning signals that should allow for learning better *image* representations as well, and recent works (Gordon et al., 2020; Alayrac et al., 2020; Wu & Wang, 2021; Tschannen et al., 2020; Xu & Wang, 2021; Jabri et al., 2020; Bian et al., 2022; Xiong et al., 2021) have made attempts in this direction. One family of successful, recent methods, use cycle-consistency-based objectives that encourage learning correspondences between temporally ordered image patches via graph random walks (Jabri et al., 2020; Bian et al., 2022). However, this is generally computationally expensive to train, and has had noted issues scaling to larger model architectures Xu & Wang (2021). Alternate approaches have used optical flow to supervise correspondence learning Sharma et al. (2022); Xiong et al. (2021), which can be quite powerful, but also limited to capturing dynamics over short time-scales. The most similar works to ours are Gordon et al. (2020), Xu & Wang (2021), and Wu & Wang (2021), which utilize simple variants of contrastive learning to learn global frame-level representations from videos. Our approach differs in its curation of video datasets and its ability to handle temporal deformations in the contrastive learning framework via learned attention. Although these works report gains on video-centric tasks such as object tracking and video segmentation, in the context of canonical scene understanding tasks used to evaluate image representations, we find these methods to underperform state-of-the-art ImageNet-pretrained models.

**Contrastive learning for fine-grained scene understanding.** In this work, we specifically focus on evaluations that assess real-world scene understanding, namely semantic segmentation and object detection (Van Gansbeke et al., 2021; Hénaff et al., 2021; Xie et al., 2021a). Self-supervised learning has greatly benefited fine-grained scene understanding tasks, and there has been significant progress using dense contrastive losses that chose positive pairs for local features by spatial proximity and/or feature affinity across two views (Xie et al., 2021b; Bai et al., 2022; O Pinheiro et al., 2020; Wang et al., 2021b). However, as described in Sharma et al. (2022), dense contrastive losses fail when ground truth correspondences cannot be easily obtained across views, as is the case when

including natural temporal augmentations. Sharma et al. (2022) propose to resolve this by tracking local features via optical flow, but these computations can be brittle and we therefore ask whether correspondences can be established using semantic similarity instead. In this work, we revert to the standard, global contrastive loss formulation, and find that it discovers semantic correspondences when equipped with a lightweight attentional module.

## 3 METHODS

In our experiments we pretrain image representations using image or video datasets, then transfer them to a range of downstream image tasks that test their spatial understanding. We adopt the ResNet-50 architecture used throughout the self-supervised learning literature.

### 3.1 PRETRAINING VISUAL REPRESENTATIONS

We start by describing our self-supervised baseline for learning representations from images or individual video frames, **MoCLR**, before adding the simple modifications which together make our method for distilling **vi**deos in**to** image representations, which we call **VITO**.

**A strong contrastive baseline.** MoCLR (Tian et al., 2021) is a simple but powerful way of learning image representations from image data. Given an image $x$ (or a frame from a video), we generate a small number of "views" via random cropping, resizing, and color jittering. Each view $v^l$ is encoded with a feature extractor $f_\theta$ into a spatial map of hidden vectors $h_\theta^l = f_\theta(v^l)$ where $\theta$ are the parameters of the *online* network being optimized. Following Chen et al. (2020), these hidden vectors are average pooled into a single vector $\hat{h}_\theta^l$ then transformed with a two-layer MLP $g_\theta$, yielding non-linear projections $z_\theta^l = g_\theta(\hat{h}_\theta^l)$ which we rescale such that their Euclidean norm is equal to $1/\sqrt{\tau}$, where $\tau = 0.1$.

We wish to enforce invariance of these features across views. In theory, one could regress one projection $z_\theta^i$ onto its target $z_\theta^j$, however it is helpful to stabilize these targets by encoding them instead with specific *target* networks $f_\xi$ and $g_\xi$, whose parameters $\xi$ vary more slowly, as shown by Grill et al. (2020). We enforce this invariance using the standard contrastive loss of van den Oord et al. (2018)

$$\mathcal{L}^{ij}(\theta; \xi) = -\log \frac{\exp(z_\theta^i \cdot z_\xi^j)}{\exp(z_\theta^i \cdot z_\xi^j) + \sum_n \exp(z_\theta^i \cdot z_\xi^n)}, \tag{1}$$

where $\{z_\xi^n\}_n$ are *negative* features from other images in the batch. We generalize this loss to multiple views by evaluating it for all pairs $\mathcal{L}(\theta; \xi) = \sum_{i \neq j} \mathcal{L}^{ij}(\theta; \xi)$. We update the online network with gradients from the contrastive loss, and the target network as an exponential moving average of the online network

$$\theta \leftarrow \text{optimizer}(\theta, \nabla_\theta \mathcal{L}(\theta; \xi), \lambda_\theta) \tag{2}$$
$$\xi \leftarrow (1 - \lambda_\xi)\xi + \lambda_\xi\theta, \tag{3}$$

where $\lambda_\theta$ and $\lambda_\xi$ are learning rates for the online and target networks respectively. By combining the contrastive formulation of SimCLR (Chen et al., 2020) and the momentum architecture of MoCo (He et al., 2020) and BYOL (Grill et al., 2020), MoCLR (similarly to MoCo v3 (Chen et al., 2021), which it is akin to) benefits from the best of each approach and has been shown to yield state-of-the-art performance on a variety of downstream tasks (Tian et al., 2021).

**Adapting synthetic augmentations to video frames.** The synthetic augmentation pipeline that has become ubiquitous in the self-supervised learning literature is tailored to pretraining models on ImageNet (Chen et al., 2020; Grill et al., 2020). However, video frames (or even uncurated image data) typically differ from the statistics of ImageNet images. In particular, uncurated video frames generally have more variable viewpoints, and a larger field-of-view that can cover multiple (not necessarily centered) objects in complex scenes. The standard random resized crop (RRC) operation, which has been found to be essential in self-supervised methods like SimCLR and BYOL (Chen et al., 2020; Grill et al., 2020), is an aggressive scale transformation where the smallest crops can cover only 8% of the original image. While this enables learning strong invariances across views when the image is relatively homogeneous in content (e.g. featuring a single object), for

video frames this can result in views that have very different semantic content (e.g. entirely different objects), dampening the selectivity of the representation for different object classes. As a result, we suggest and will empirically validate that larger crop sizes (e.g. increasing the minimum crop size to 40%) are beneficial when applying contrastive learning to video frames.

**Learning from natural temporal transformations.** When learning from still images, we apply the random pre-processing pipeline of BYOL (Grill et al., 2020), which includes random cropping, flipping, blurring, and point-wise color transformations $v^l \sim \mathcal{A}_l(x)$, see appendix A.1 for the the detail of their distributions. When learning from videos, we sample frames according to a distribution $\mathcal{T}$ and transform each frame using the same pipeline as above:

$$v^1 \sim \mathcal{A}_1(x_1) \qquad v^2 \sim \mathcal{A}_2(x_2) \qquad x_1, x_2 \sim \mathcal{T}(\{x_t\}_{t=1,...,T}) \tag{4}$$

Recent works have suggested similar methodologies for learning from natural augmentations. In Gordon et al. (2020), $\mathcal{T}$ samples pairs with a fixed time delay. Xu & Wang (2021) choose a distribution that involves uniform sampling over independent chunks of a given video clip. In this work, we propose a simpler approach where we sample from a uniform distribution over the entire video segment of length $T = 2.56s$. In Figure A.2, we show that these sampling schemes induce very different distributions of absolute time differences between pairs of frames. Our marginal sampling scheme is arguably the most natural as the mode of the distribution is at 0, meaning that it is not biased to over-represent any specific time difference (similarly to the random-resized crop operation in space). While not a huge effect, we find that when evaluated on multiple downstream tasks, this temporal sampling method outperforms other methods (Figure A.2).

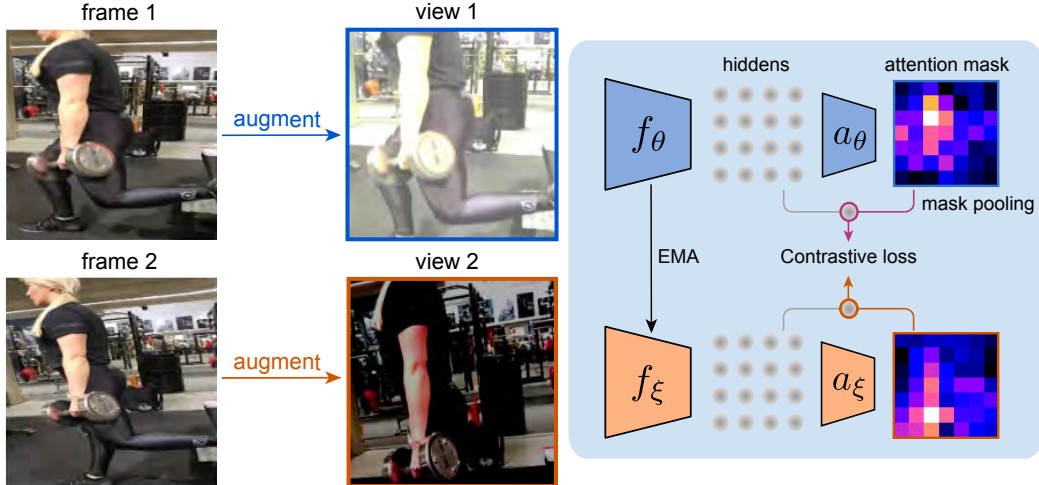

Figure 2: Learning to attend to related video content. Each augmented frame is encoded by the network $f$ as a spatial array of hidden vectors. The attention module $a$ takes as input features from one view and produces a mask that isolates features that are likely to be predictive of the other, temporally-displaced view. The attention-gated features are pooled accordingly, and both the feature extractor and attention module are trained to satisfy the contrastive objective. Subscripts $\theta$ and $\xi$ refer to online and target (EMA) networks respectively.

**Multi-scale contrastive attention pooling.** Typical contrastive frameworks use a simple global average pooling of hidden vectors to obtain a single feature for each input that can then be passed through the projector network $g_\theta$. This global average pooling (GAP) aggregates content across the whole image, which can enable higher-level semantic tasks like image classification. However, global pooling makes it hard to localize features in space. As a result, for fine-grained tasks like semantic segmentation, it has been shown that using dense contrastive losses can lead to significant improvements (Wang et al., 2021b; Xie et al., 2021b; Bai et al., 2022; Hénaff et al., 2021).

Most local contrastive methods require establishing local correspondences across the two views of the input image such that the contrastive loss can be applied to features that represent the same image content. While these correspondences easily obtained when learning from static images, when temporal deformations are introduced they require some form of object or point tracking (Sharma et al.,

2022). Yet these methods can be quite cumbersome and involve tuning many dataset-dependent hyperparameters, so in this work, we propose a more general, adaptive method for learning (at multiple scales) what features should be attended to in order to solve the contrastive learning problem across temporally displaced views.

As shown in Figure 2, given a view $v^l$ the feature extractor outputs a spatial map of feature vectors $h_\theta^{l,s} \in \mathcal{R}^{h \times w \times c}$ at a given scale $s$, where different scales correspond to the outputs of different blocks of a ResNet for example. At each scale, we introduce a 2-layer attention MLP $a_\theta^s$ which outputs a mask $m^{l,s} = softmax(a_\theta(h_\theta^{l,s}))$ that we use to spatially weight and pool hidden vectors:

$$\hat{h}_\theta^{l,s} = \frac{1}{\sum_{i,j} m^{l,s}[i,j]} \sum_{i,j} m^{l,s}[i,j] \, h_\theta^{l,s}[i,j]. \tag{5}$$

Given the attention-pooled features from multiple scales, we concatenate them before transforming them with the two-layer MLP projector: $z_\theta^l = g_\theta(\hat{h}_\theta^l)$ where $\hat{h}_\theta^l = [\hat{h}_\theta^{l,s}, \; s \in 1...S]$.

This framework allows the projector to utilize a multi-scale, adaptively localized representation to solve the contrastive learning problem. This is especially important given the much larger dynamic range of scales at which objects can appear in videos as opposed to single-object ImageNet images. This method is related to that of Jetley et al. (2018), which applied a variant of spatial attention pooling in the context of supervised image classification, and more loosely to attention-based backbones which have shown great success in self-supervised learning (Caron et al., 2021). Note however that, rather than requiring specialized network operations, our multi-scale attention pooling module can be a lightweight addition to standard convolutional architectures. In our experiments, we find that for the canonical ResNet-50 architecture, attending over the outputs of the last two ResNet blocks (i.e. $S = 2$) is optimal given our evaluations.

## 3.2 EVALUATING VISUAL REPRESENTATIONS

Classification has traditionally been the default means of evaluating image representations. Classifying single objects however does not require many of the defining features of real-world scene understanding. Semantic segmentation and object detection provide more relevant tests as they require that a representation understand fine and coarse object boundaries, shapes, sizes, and viewpoints. We therefore evaluate on two semantic segmentation datasets, PASCAL VOC (Everingham et al., 2015) and ADE20K (Zhou et al., 2017), which respectively test object-level understanding and complex scene understanding. We also evaluate on the well-known COCO object detection dataset (Lin et al., 2014) and the more challenging long-tailed LVIS dataset (Gupta et al., 2019). To test generalization beyond image benchmarks, we also evaluate on two video-based tasks, DAVIS 2017 Segmentation and UCF-101 action recognition. For details on the specific training and evaluation protocols see Sec A.2.

## 3.3 ADDRESSING DATASET DOMAIN MISMATCH

We begin investigating the potential for video learning with standard datasets including Kinetics, AudioSet, and YouTube-8M. Yet prior work has shown that even self-supervised methods are sensitive to the pretraining distribution. We therefore hypothesized that video pretraining might benefit from a data distribution that is more aligned with the statistics of standard image datasets.

As a test of this hypothesis, we developed a simple data curation pipeline (which we refer to as *VideoNet*) to filter online videos such that our training data more closely matches the distribution of ImageNet categories. For each of the 1,000 ImageNet categories, we retrieved 5,000 video clips whose title included the category's name or a synonym. We then filtered these videos by applying an image classifier to verify that the videos contained the intended object category. For this we ran a pretrained ResNet-50 ImageNet classifier on the first 100 frames of each video and discarded videos for which the query category was not equal to the ResNet's top-1 prediction for any of the frames. We additionally discarded videos of less than $10s$ in length. This procedure resulted in a dataset of 1,180,042 videos in total.

We note that while the VideoNet procedure is close in conceptualization to the method used to create the R2V2 dataset proposed by Gordon et al. (2020), it differs in a few ways. First, we utilize full

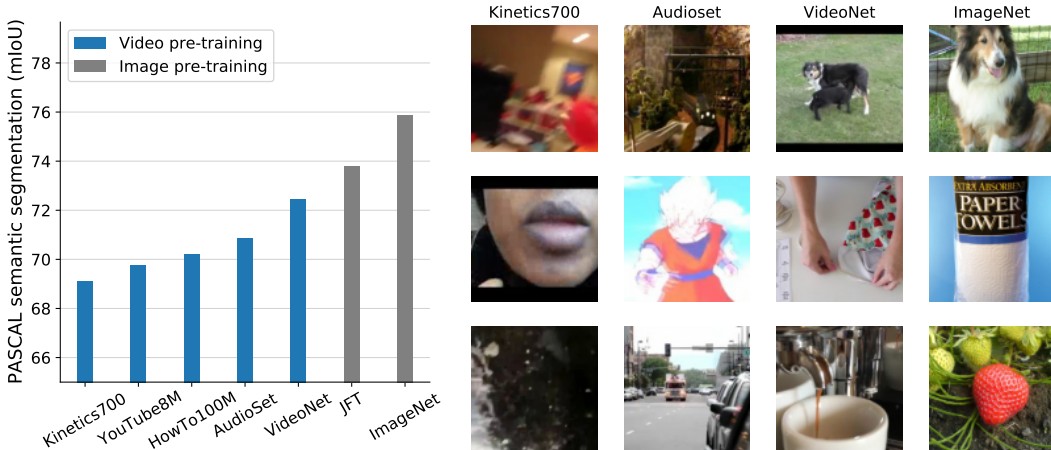

Figure 3: Impact of pretraining data's spatial content on representation quality. Left: transfer performance of models pretrained on single frames from image datasets (grey bars) or individual videos (blue bars). All models use a ResNet-50 backbone and 2-view MoCLR pretraining for 300 epochs. Right: example frames from different video and image dataests.

video clips that allow us to uniformly sample frames at any time point rather than the fixed sampling of frames that are $5s$ apart in R2V2. This coarse temporal sampling reduces the total number of frames a model can learn from, but also limits the resolution of temporal deformations used in the contrastive framework: displacements of $5s$ are more likely to have changes in semantic content than continuously sampled frames from a small interval. Second, by using the ImageNet classifier to filter videos, we can reduce mismatch with the ImageNet distribution that can arise from incorrect tagging and noisy labeling of online videos. This is somewhat verified by the fact that only 24% of the retrieved videos met our filtering criteria.

## 4 RESULTS

### 4.1 EFFECT OF PRETRAINING DATA

To demonstrate the effect of the pretraining data distribution on transfer performance, we first pretrain the baseline MoCLR model (using 2 views) on a variety of image and video datasets, where we initially treat video datasets as collections of individual frames. We train each model for 300 ImageNet-equivalent epochs, referred to hereafter as "epochs" (i.e. 1 epoch = learning from 1.28M examples, irrespective of the dataset), such that each model benefits from the same amount of computation. Figure 3 (left) shows their transfer performance on PASCAL semantic segmentation.

**Training on standard datasets.** As expected, ImageNet pretraining works very well, but pretraining on standard video datasets results in a substantial drop in performance (e.g. $-6.8\%$ or $-5\%$ mIoU from pretraining on Kinetics700 or AudioSet). This performance gap between video and image pretraining can be attributed to a combination of increased complexity and field-of-view of video frames and domain mismatch between the dataset categories (Figure 3, right). Consistent with this, training on JFT (Sun et al., 2017), an uncurated dataset with a heavy-tailed class distribution, also results in a loss in performance. Notably, this is despite the much larger size of JFT (300M images), which indicates that *having more training data does not necessarily lead to increased performance*.

**Training with VideoNet curation.** We find that applying the same baseline pretraining to frames from our curated video dataset performs better than existing large-scale video datasets like Audioset ($+1.6\%$ mIoU), but still underperforms image pretraining on JFT and ImageNet (Figure 3).

This result demonstrates the importance of aligning the image frame distributions between video and image datasets. As a result, we choose this filtered video dataset as our primary pretraining dataset for the rest of this work, with the goal of closing the gap with ImageNet pretraining performance.

## 4.2 Leveraging video data with VITO

We now turn to our proposed method for distilling videos into image representations, **VITO**. We investigate the choices in our pretraining paradigm by varying each of the three main components in isolation: adaptation of spatial crop size, natural augmentations, and multi-scale attention pooling. For the following ablations, we train VITO and its variants for 300 epochs on VideoNet, using 3 views, then transfer to PASCAL semantic segmentation. Similarly, we pretrain MoCLR baselines on ImageNet and VideoNet for 300 epochs using 3 views.

**Adapting spatial augmentations.** We validate in Figure 4 (left) our hypothesis that increasing the minimum crop-scale in the random-resized crop operation during training leads to models that generalize better to fine-grained tasks like semantic segmentation. Specifically, we find that a minimum crop scale of 0.4 (as opposed to the traditional 0.08) results in the best transfer performance ($+1.7\%$ mIoU). Note that this conclusion differs slightly from that of Feichtenhofer et al. (2021) who find more aggressive cropping to be beneficial for action recognition.

**Natural augmentations.** As described in section 3.1, for each training example, we sample 3 views using marginal sampling of each frame from the video clip of length $T = 2.56$ seconds. This length determines the distribution (and accordingly the mean) time difference between any pair of frames. As a result, the total length impacts the time-scale over which the contrastive model learns invariances. We verify our choice by varying the total length of clips. While going to longer time-scales $T = 3.2s$ does not hurt performance much, we find a significant improvement over using shorter clips (e.g. $T = 1.28s$, $+1.0\%$ mIoU; Figure 4, center). This suggests that invariance to the rich temporal deformations present in video clips is indeed a beneficial criterion for learning fine-grained spatial representations. Note however that the optimal temporal displacement is relatively short (median $= 0.76s$ when $T = 2.56s$, Figure A.2) and that sampling video datasets too coarsely (e.g. every $5s$ as in Gordon et al. (2020)) may therefore limit their utility.

**Multi-scale attention pooling.** We decompose the proposed multi-scale contrastive attention pooling to isolate the effects of multi-scale learning from those of attention pooling. While we find only modest gains from adding attention pooling to a single-scale version of the model ($+0.2\%$ mIoU), we find that the 2-scale model (without attention pooling) improves over the single scale model more robustly ($+0.6\%$ mIoU). Interestingly, we find that the combination of the 2-scale model with attention pooling has a synergistic effect ($+1\%$ mIoU over the single-scale attention model), highlighting the importance of handling the variability in scales present in natural videos. Furthermore, through visualization of the attention masks, we also discover an interesting property of semantic-binding that we believe underlies these performance gains (see Sec A.3 for more discussion).

**Combined**, the three modifications VITO makes to the contrastive framework result in a $2.8\%$ mIoU improvement over MoCLR pretrained on VideoNet, closing the gap with ImageNet pretraining when transferring to PASCAL semantic segmentation. In the next sections, we seek to understand the mechanism underlying this improvement, and validate it on other downstream tasks.

## 4.3 Evaluation: image understanding tasks

Having shown that VITO is potentially learning novel visual representations that transfer well to PASCAL segmentation, we present in Table 1 the transfer performance of VITO against recent image and video pretraining methods on all of the semantic segmentation and object detection tasks.

**Comparison to video-pretraining.** Given a similar computational budget as prior works (200 epochs and 3 views) VITO delivers large gains over all prior methods. For example, VITO improves over VIVI (Tschannen et al., 2020) by 10%/5%/2%/2%, highlighting the importance of data curation and our contrastive formulation. VITO improves over VINCE (Gordon et al., 2020) by 7%/4%/1%/1%, highlighting the importance of fine-grained temporal deformations. Finally, VITO improves even over MMV (Alayrac et al., 2020) by 5%/7%/2%/2%, despite their use of large-scale text supervision, highlighting the relevance of video-only learning. Finally, we disentangle the power of our method and dataset by confirming that each independently have strong effects: MoCLR trained on VideoNet still outperforms all prior work. Similarly VITO trained on standard datasets (Audioset or YT8M) also outperform all prior work.

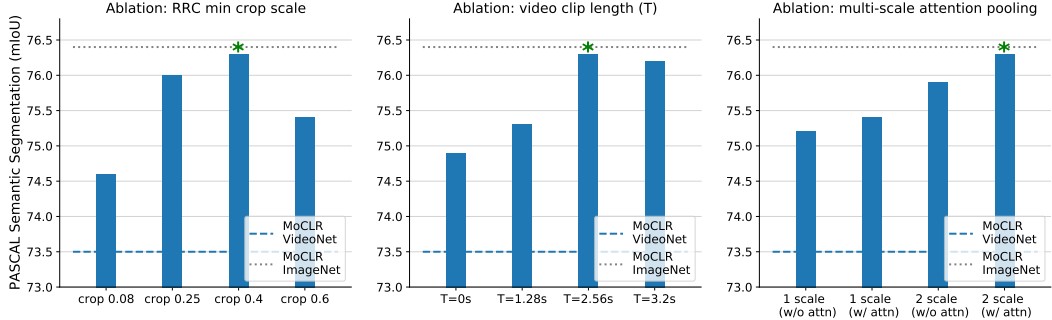

Figure 4: Effects of crop scale, natural augmentations, and multi-scale attention on representation quality. All ablations are performed relative to VITO's configuration (denoted by a green asterisk) which uses 2-scale attention pooling, a less aggressive crop scale of 40%, and natural augmentations uniformly sampled in a window of length T = 2.56s. We also compare to our baseline MoCLR model trained on single frames, either from ImageNet (dotted gray line) or VideoNet (dashed blue line). All models are evaluated by transferring to PASCAL semantic segmentation.

| Pretraining | Dataset | Epochs | Semantic segmentation | | Object detection | |
|---|---|---|---|---|---|---|
| | | | PASCAL | ADE20K | COCO | LVIS |
| Random Init | | | 53.0 | 27.9 | 39.0 | 21.1 |
| *Comparison to video pretraining* | | | | | | |
| VFS (Xu & Wang, 2021) | K400 | 100 | 63.9 | 31.4 | 41.6 | 23.2 |
| VIVI (Tschannen et al., 2020) | YouTube8M | 192 | 65.8 | 34.2 | 41.3 | 23.2 |
| VINCE (Gordon et al., 2020) | R2V2 | 200 | 69.0 | 35.7 | 42.4 | 24.4 |
| CycleContrast (Wu & Wang, 2021) | R2V2 | 200 | 69.2 | 35.6 | 42.8 | 24.5 |
| MMV (Alayrac et al., 2020) | AS + HT | 1600 | 70.6 | 32.5 | 41.3 | 24.2 |
| MoCLR | VideoNet | 200 | 72.8 | 37.5 | 42.6 | 24.6 |
| VITO | YT8M | 200 | 71.8 | 37.8 | 42.7 | 24.6 |
| VITO | AudioSet | 200 | 73.6 | 38.5 | 43.2 | 25.0 |
| VITO | VideoNet | 200 | **75.5** | **39.2** | **43.6** | **25.6** |
| | | | | | | |
| *Comparison to ImageNet pretraining* | | | | | | |
| Supervised | ImageNet | 200 | 71.3 | 33.5 | 44.2 | 25.2 |
| BYOL (Grill et al., 2020) | ImageNet | 300 | 76.1 | 38.8 | 43.7 | 25.5 |
| MoCLR (Tian et al., 2021) | ImageNet | 300 | 76.4 | 39.2 | 43.9 | 25.8 |
| DINO (Caron et al., 2021) | ImageNet | 300 | 76.1 | 39.0 | 44.3 | 26.4 |
| VITO | VideoNet | 300 | **76.3** | **39.4** | **44.0** | **25.7** |

Table 1: VITO outperforms prior video pretraining and closes the gap with ImageNet-pretraining of ResNet-50 models. For external models, we finetune publicly available checkpoints.

**Comparison to ImageNet pretraining.** Finally, we compare our VITO-pretrained VideoNet model to a host of state-of-the-art ImageNet-pretrained methods. Surprisingly, we find VITO to be competitive with the best of such methods, outperforming BYOL and DINO on PASCAL, matching BYOL and MoCLR on COCO and LVIS, and surpassing all methods on ADE20K. VITO largely surpasses supervised ImageNet pretraining on 3 downstream evaluations. This is, to the best of our knowledge, the first example of video pretraining achievingImageNet-level performance on such tasks. We also demonstrate that our results are not specific to the ResNet-50 architecture and scale well when using Swin transformers (see Sec. A.4).

### 4.4 EVALUATION: VIDEO UNDERSTANDING TASKS

To further emphasize the value of video pretraining, we evaluated VITO on tasks that have been proven difficult when using image pretraining, specifically video segmentation and action recognition. We find these tasks to be a good combination as they test both 1-shot fine-grained capabilities (DAVIS segmentation) and coarser temporal understanding (UCF101 full video classification).

| Pretraining | Dataset | $\mathcal{J}_m$ | $\mathcal{F}_m$ |
|---|---|---|---|
| DetCon$_B$ (Hénaff et al., 2021) | ImageNet | 63.1 | 66.4 |
| MoCLR (Tian et al., 2021) | ImageNet | 63.1 | 67.8 |
| BYOL (Grill et al., 2020) | ImageNet | 63.8 | 69.4 |
| VITO | VideoNet | **65.5** | **70.8** |

Table 2: VITO outperforms all image-pretraining baselines on DAVIS 2017 video segmentation

| Pretraining | Dataset | Backbone | Top-1 |
|---|---|---|---|
| *Video architectures* | | | |
| Supervised (Wang et al., 2021a) | ImageNet | I3D | 67.1 |
| VideoMoCo (Pan et al., 2021) | K400 | R(2+1)D | 78.7 |
| Temporal-ssl (Jenni et al., 2020) | K400 | R(2+1)D | 81.6 |
| VTHCL (Yang et al., 2020) | K400 | 3D-R50 | 82.1 |
| CVRL (Qian et al., 2021) | K400 | 3D-R50 | 92.9 |
| $\rho$-BYOL (Feichtenhofer et al., 2021) | K400 | 3D-R50 | 95.5 |
| *Image architectures* | | | |
| Shuffle and Learn (Misra et al., 2016) | UCF101/HMDB51 | AlexNet | 50.6 |
| OPN (Lee et al., 2017) | UCF101 | VGG-M | 59.8 |
| TCE (Knights et al., 2021) | K400 | R50 | 71.2 |
| CycleContrast (Wu & Wang, 2021) | R2V2 | R18 | 76.8 |
| CycleContrast (Wu & Wang, 2021) | R2V2 | R50 | 82.1 |
| VITO | VideoNet | R50 | **85.3** |

Table 3: VITO outperforms all 2d pretained backbones in finetuning on UCF101 w/ just pooling of frame representations. Performance is even above all video architectures except gray numbers.

**DAVIS segmentation.** Here we demonstrate the value of video pretraining compared directly with ImageNet pretraining. In addition to displaying competitive scene understanding capabilities, we see that VITO specifically learns features relevant to temporal dynamics, and conclusively outperforms strong ImageNet pretraining methods, on both the region jaccard ($\mathcal{J}$) and boundary F measure ($\mathcal{F}$) (Table 2).

**UCF101 action recognition.** In Table 3, we present finetuned top-1 accuracy of many methods that are either pretrained on video data using video-specific architectures (3D convolutions and variants) or standard image architectures that produce frame-based representations which are simply averaged over clips. Traditionally, the latter have not been able to come close to the performance of the former, but we find that VITO has the best performance of all image backbones and even outperforms many recent video-specific SSL methods.

## 5 DISCUSSION

We propose VITO, a simple method for distilling videos into image representations. The key features of our method include improved dataset curation, adapting standard synthetic augmentations to video frames, and using attention-guided contrastive learning. With these components, VITO outperforms all prior video pretraining methods on object detection and semantic segmentation tasks, and for the first time, closes the gap with ImageNet pretraining. Furthermore, unlike image pretraining, VITO-pretraining generalizes to tasks that require temporal understanding, achieving surprisingly strong performance on video segmentation and action recognition.

We believe this work can be a foundation for future video pretraining efforts, as our approach is powerful, yet simple and extensible in almost every aspect. For example, because we base our learning paradigm on standard architectures and contrastive learning methods, it is easy to extend or adapt our approach to leverage continuing advancements in image-based contrastive, and more generally, self-supervised learning objectives. Additionally, while we have shown the benefits of a simple attention module for learning from video data, there are great opportunities to extend our approach to take advantage of more powerful attentional architectures. In sum, despite the many successes in video representation learning, our results suggest that there is a great untapped potential in video pretraining as a paradigm for learning general visual representations.

## 6 REPRODUCIBILITY STATEMENT

We will release our pretrained models along with the code needed to implement the VITO model architecture. Pretraining and evaluation details about architectures, optimization, and hyperparameters have all been detailed in the appendix. The VideoNet procedure for curating video datasets can be reproduced with standard ImageNet classifiers and publicly available online videos.

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

# A  APPENDIX

## A.1  IMPLEMENTATION: DATA PRE-PROCESSING

**Self-supervised pretraining.** Each frame is randomly augmented by composing the following operations, each applied with a given probability:

1. random cropping: a random patch of the image is selected, whose area is uniformly sampled in $[s \cdot \mathcal{A}, \mathcal{A}]$, where $\mathcal{A}$ is the area of the original image, and whose aspect ratio is logarithmically sampled in $[3/4, 4/3]$. $s$ is a scale hyper-parameter set to $0.08$ when learning from ImageNet, and $0.4$ when learning from videos. Regardless, the patch is then resized to $224 \times 224$ pixels using bicubic interpolation;

2. horizontal flipping;

3. color jittering: the brightness, contrast, saturation and hue are shifted by a uniformly distributed offset;

4. color dropping: the RGB image is replaced by its grey-scale values;

5. gaussian blurring with a $23 \times 23$ square kernel and a standard deviation uniformly sampled from $[0.1, 2.0]$;

6. solarization: a point-wise color transformation $x \mapsto x \cdot \mathbb{1}_{x<0.5} + (1 - x) \cdot \mathbb{1}_{x \geq 0.5}$ with pixels $x$ in $[0, 1]$.

The augmented frames $\boldsymbol{v}^1$ and $\boldsymbol{v}^2$ result from augmentations sampled from distributions $\mathcal{A}_1$ and $\mathcal{A}_2$ respectively. These distributions apply the primitives described above with different probabilities, and different magnitudes. The following table specifies these parameters for the BYOL framework (Grill et al., 2020), which we adopt without modification. When learning from three views, we use the distribution $\mathcal{A}_1$ to generate the third view.

| Parameter | $\mathcal{A}_1$ | $\mathcal{A}_2$ |
|---|---|---|
| Random crop probability | 1.0 | |
| Flip probability | 0.5 | |
| Color jittering probability | 0.8 | |
| Color dropping probability | 0.2 | |
| Brightness adjustment max | 0.4 | |
| Contrast adjustment max | 0.4 | |
| Saturation adjustment max | 0.2 | |
| Hue adjustment max | 0.1 | |
| Gaussian blurring probability | 1.0 | 0.1 |
| Solarization probability | 0.0 | 0.2 |

**Transfer to PASCAL and ADE20K.** During training, images are randomly flipped and scaled by a factor in $[0.5, 2.0]$. Training and testing are performed with $512 \times 512$-resolution images. When fine-tuning on ADE20K, we aditionally use photometric transformations from the mmseg[1] codebase.

**Transfer to COCO and LVIS.** The target resolution is $800 \times 1024$. During testing, an image is resized by a factor $s$ while preserving the aspect ratio, such that it is tightly contained inside the target resolution, and then padded. When fine-tuning, the image is rescaled by a factor of $u \cdot s$ where $u$ is uniformly sampled in $[0.8, 1.25]$, and is then cropped or padded to the target resolution.

## A.2  IMPLEMENTATION: OPTIMIZATION

**Self-supervised pretraining.** We pretrain ResNet-50 using the LARS optimizer (You et al., 2017) with a batch size of 4096 split across 128 Cloud TPU v3 workers. We adopt the optimization details of BYOL, scaling the learning rate linearly with the batch size and decaying it according to a cosine schedule. The base learning rate is 0.3 and the weight decay is $10^{-6}$.

---

[1] https://github.com/open-mmlab/mmsegmentation

**Semantic segmentation on PASCAL and ADE20K.** We evaluate ResNet models by attaching a fully-convolutional network (FCN, Long et al. (2015)) and fine-tuning end-to-end, following He et al. (2020). We fine-tune for 45 and 60 epochs on PASCAL and ADE20K respectively, and report the mean intersection over union (mIoU) averaged across 5 runs.

We fine-tune for 45 epochs on the PASCAL `train_aug2012` set or 60 epochs on the ADE20K `train` set. We use stochastic gradient descent with a batch size of 16 and weight decay of 0.005. The learning rate is initially set to 0.04 and decayed exponentially with a factor of $0.9^n$ where n is the iteration number. When fine-tuning external models, we sweep over the base learning rate and weight decay and report their performance given the optimal configuration. In all cases we report mIoU on the `val` set averaged across 5 runs.

**Transfer to COCO and LVIS with FCOS$^\star$.** The network is fine-tuned for 30 epochs on the COCO `train2017` set or the LVIS `v1_train` set. We use AdamW (Loshchilov & Hutter, 2019) with weight decay $10^{-4}$, base learning rate of $10^{-3}$, and batch size 128 split across 16 workers. The learning rate rises linearly for $\frac{1}{4}$ of an epoch, and is dropped twice by a factor of 10, after $\frac{2}{3}$ and $\frac{8}{9}$ of the total training time. We report mAP on the COCO `val2017` set and the LVIS `v1_val` set, averaged across 5 runs.

We evaluate pretrained ResNet's using the FCOS$^\star$ architecture, following Hénaff et al. (2022). FCOS$^\star$ is the implementation of a single-stage detector based on FCOS (Tian et al., 2019), and improved with the collection of techniques from Wu et al. (2020), Zhang et al. (2020), and Feng et al. (2021), full details can be found in Hénaff et al. (2022). The pretrained network is used to initialize the backbone of the FCOS$^\star$ model, which is then fine-tuned for 30 epochs. We report bounding-box mean average precision (mAP) averaged across 5 runs.

**Video segmentation on DAVIS**

As a further test of scene understanding, we assess whether learned representations can continue to recognize parts of an object as they evolve over time. Video object segmentation, specifically in its semi-supervised setting, captures this ability, which we evaluate on the DAVIS'17 benchmark. Having evaluated a learned representation on a video independently across frames, we segment these features with nearest neighbor matching from frame to frame, given a segmentation of the first frame. In this way, the segmentation is propagated according to the similarity of the representation across space and time. We reuse the segmentation procedure from Xu & Wang (2021) without modification.

**Action recognition on UCF101** We evaluate action recognition classification on the UCF101 dataset (Soomro et al., 2012). We follow the procedure for finetuning used in Wu & Wang (2021) which is based on Morgado et al. (2021). Clip representations are obtained by averaging the frame representations for the video, and one fully connected layer is used for predicting the action class. 10 clips are sampled from each video and the predictions of the clips are averaged for the final results.

## A.3 Semantic binding with contrastive attention pooling

The ablation study demonstrated that multi-scale attention improves the performance of VITO in semantic segmentation. To probe why this may be, we visualize and interpret the learned attention masks (Figure A.1). For simplicity, we only visualize the masks from the coarsest scale (output feature map), but the interpretation naturally extends to the multi-scale version as these masks are learned with independent attention modules.

Because the attention masks are not computed jointly across each view, for a given video frame, the attention module must marginalize over the training data to make a statistical prediction—what should be attended to in the first view in order to minimize the contrastive loss across possible second views? Specifically, the attention must focus on content that is most likely to be stable across time while still being discriminative (or unique) relative to other frames from other videos. Different examples appear to trade-off these criteria differently, yet systematically. For example, in the third column of Figure A.1 even though the animated characters on the right side of both frames may be discriminative content, the attention module has learned to focus on the static picture on the left as it is the content that is most likely to be stable across time. For this pair of frames the prediction is

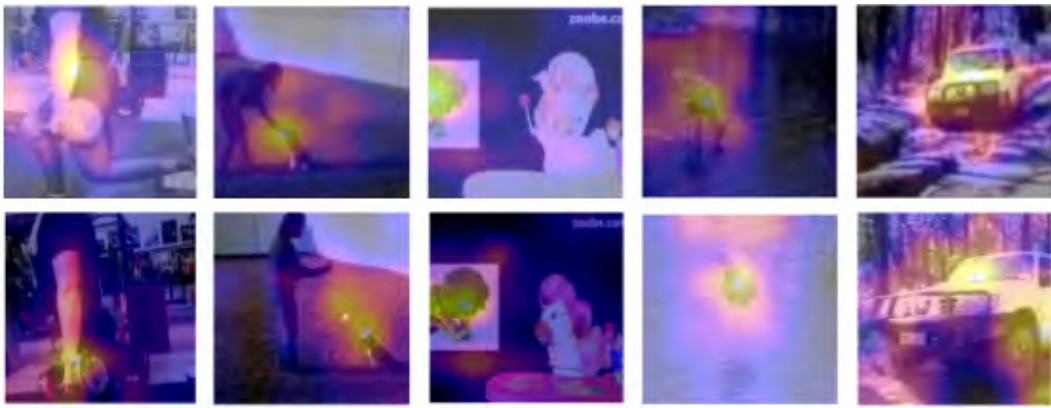

Figure A.1: Example augmented frames with overlaid (resized) learned attention masks. Attention is computed from the output of the final block of the VITO trained ResNet-50. Crucially, the attention masks are computed independently, such that the attention module can only use spatial cues.

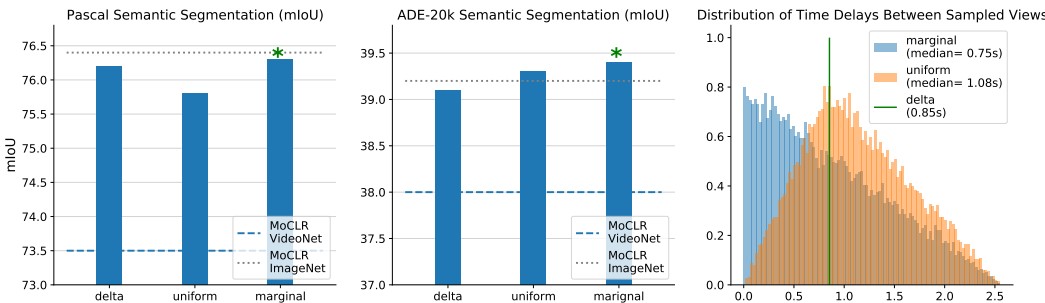

Figure A.2: Ablating different temporal sampling schemes. *Delta* refers to fixed time sampling between frames as in Gordon et al. (2020). *Uniform* refers to chunking time into non-overlapping blocks and uniformly sampling within each chunk as in Xu & Wang (2021). *Marginal* sampling (ours) refers to simple uniform sampling from the full video clip of length $T = 2.56s$. First two panels show that marginal sampling is best overall across transfer to PASCAL and ADE20K. Third panel shows the distribution of absolute time-differences between any two pairs of frames under each sampling scheme (assuming 3 views are sampled per clip).

correct—the attention disregards content that is changing too abruptly—despite not having access to motion cues. On the other hand, the example in the fourth column demonstrates a scenario where the model has attended to stable, but primarily discriminative content (the bird) rather than the background, which is also very stable but most likely less unique relative to other videos.

Even beyond the ability to localize stable, yet discriminative content, it seems that our method also enables "semantic binding" of visually different, but semantically related features. This can be seen in the first pair of frames, as the model has learned to associate an arm or elbow (in the first frame) with the dumbbell (in the second frame), demonstrating an understanding that these two semantically related concepts co-occur and thus are predictive of one another given the right embedding.

Binding co-occuring features appears as an intuitive explanation for why these representations would perform well on semantic segmentation. It is particularly interesting that training end-to-end with a standard contrastive loss can produce complex behavior reminiscent of the DINO approach (Caron et al., 2021) even though we use a single, two-layer MLP attention module as opposed to large-scale transformer architectures which use attention throughout the network.

## A.4    MODEL SCALING RESULTS

| Pretraining | Dataset | Backbone | Semantic segmentation | | Object detection | |
|---|---|---|---|---|---|---|
| | | | PASCAL | ADE20K | COCO | LVIS |
| VITO | ImageNet | R50 | 76.3 | 39.4 | 44.0 | 25.7 |
| MoCLR (Tian et al., 2021) | ImageNet | Swin-S | 78.6 | 43.7 | 48.4 | 32.7 |
| VITO | VideoNet | Swin-S | **81.3** | **46.1** | 49.8 | **33.5** |
| Detcon$_B$ (Hénaff et al., 2021) | ImageNet | Swin-S | **81.4** | **46.1** | **50.4** | 33.1 |

Table A.1: VITO scales to larger model architectures (Swin-S), improving performance compared to the ResNet-50 baseline and remaining competitive with ImageNet pretraining.

