# OpenReview forum: "Self-supervised video pretraining yields strong image representations"
_ICLR.cc/2023/Conference — Submitted to ICLR 2023_

### Official Review · Reviewer_zSuF · 2022-10-15

**Confidence:** 4
**Correctness:** 3
**Technical Novelty And Significance:** 2
**Empirical Novelty And Significance:** 2
**Recommendation:** 3

**Clarity, Quality, Novelty And Reproducibility:**

The paper is well written generally and the hyper parameters alongside the implementation details are listed in the appendix.

**Strength And Weaknesses:**

Strengths:

- Overall, the paper is well written
- The results confirm that with careful considerations (by filtering the data and tuning the augmentations) video based pretraining can match in certain conditions imagenet pretraining.

Weaknesses and questions:

Low technical novelty:

- the method itself is a direct adaptation/implementation of MoCLR.

- The study of adapting the augmentation to videos, while reasonable, is straight-forward and its more of a hyper-parameter tuning than an actual contribution (e.g. setting the augm scale and frame sampling).

- The data filtering pipeline addresses the distribution mismatch between ImageNet and the video dataset. It's not surprising that matching the data distribution will improve the performance and such pipelines, are common, albeit perhaps not for this very specific use case.

Insufficient evaluation, ablations and comparisons:

- As multiple frames are present in a video, counting all the frames, the filtered dataset is still 1 order of magnitude higher than imagenet. What is the performance when the same number o frames as that of the imagenet images are used?

- Expanding on the prior question, what is the performance of a model trained with a single frame (most representative according to the imagenet classifier used for filtering)? This should hopefully emphasize how much the additional dynamic present in video helps.

- No results and ablation with other backbones: how does the performance vary when the backbone is larger (e.g: ResNet-50-2x, ViT-B/L) and respectively smaller?

- Current state-of-the-art for video recognition is achieved by initializing from pre-training on static images. How does the performance vary?

- Insufficient evaluations: only object detection and segmentation considered. What is the performance on classification for example? What is the few-shot performance of such models?

- How does this compare with imagenet-22k pretraining? How would the filtering perform based on these categories?


**Summary Of The Paper:**

The paper describes a series of hyper-parameter changes (i.e. scale of the crop, frame sampling) and data filtering for learning representation from videos. The changes are implemented on top MoCLR. On the dataset and tasks tested the paper results show that video pretraining can match imagenet pretraining. However, the technical novelty is limited and the experimental results insufficient.

**Summary Of The Review:**

While the result itself is interesting, the evaluation is very limited, with insufficient ablation and comparisons. Moreover, on the technical side, the novelty is very low as the approach is a direct implementation of an existing technique with appropriate hyper parameter tuning for video data.

---

> ### Author Response · Authors · 2022-11-17
> **Specific response to reviewer zSuF**
>
> We thank the reviewer for the constructive feedback and address their concerns point-by-point (reviewer comments in italics):
>
> 1. _“Low technical novelty…”_
>
> We agree that the components of VITO are standard on their own, but believe the idea of using multi-scale attention pooling for learning correspondences across frames is novel within this context. Specifically, it is a conceptual departure from recent efforts that have tried to do explicit object tracking to extract “views” for contrastive learning [1, 2] or methods that use explicit masks for a similar purpose in spatial contrastive learning [3].
>
> In addition, while data curation has existed for years, we think that showing that it is _sufficient_ (together with the other simple changes we propose) to close the gap with ImageNet pretraining is a very valuable contribution to the field: video pretraining can now be the default way of pretraining image representations that benefit from increased temporal understanding without sacrificing spatial understanding (see Q1 in the general response).
>
> [1] Sharma, et al. "Pixel-level Correspondence for Self-Supervised Learning from Video." ICML 2022
>
> [2] Xiong et al. "Self-Supervised Representation Learning from Flow Equivariance." _ICCV 2021_
>
> [3] Hénaff et al. "Efficient visual pretraining with contrastive detection." ICCV 2021
>
> 2. _“...performance when the same number of frames as that of the imagenet images are used?”_
>
> We respectfully point out that this is a common misconception: more data does not necessarily help in self-supervised learning. We show that increasing the size of image datasets by a factor of 300x in fact hurts performance substantially. Both JFT and VideoNet have roughly 300M frames in total, yet VITO trained on VideoNet performs substantially better than MoCLR pretrained on JFT  (see Table in general response Q2). Hence, more important than the number of frames is the distribution of those data, and the learning algorithm applied to them, which we show in this work.
>
> 3. _“what is the performance of a model trained with a single frame ... how much the ... dynamics present in video helps.”_
>
> Our ablation shows that training with a single frame at a time (instead of 2s clips) hurts performance (Figure 4, middle). In addition, our results on the JFT dataset (see general response) show that performance decreases compared to ImageNet, even though the model still has access to all 300M frames. This emphasizes that the total number of frames isn’t particularly important, but that learning from the dynamics across video frames is helpful in learning strong image representations.
>
> 4. _“No results and ablation with other backbones...”_
>
> We ran VITO on VideoNet with the high-performance Swin-S backbone, and find that it scales very well with model capacity (+5%/6%/6%/8% our 4 benchmarks, see below). Importantly, it greatly outperforms MoCLR trained with the same backbone, and is competitive with the recently proposed DetCon ImageNet pretraining (Hénaff et al., ICCV 2021), which is tailored to scene understanding tasks. We will include these results in the final text.
>
> | **Method**      | **Dataset** | **Arch** |  **PASCAL**   | **ADE20K**  | **COCO**   | **LVIS**  |
> | :---        |    :----:   |    :----: | :----: |    :----:   |    :----:  |   ---: |
> | VITO        |    VideoNet   | ResNet-50 |   76.3 |   39.4 | 44.0 |  25.7 |
> | MoCLR |    VideoNet   |  Swin-S |  78.6 |    43.7 | 48.4 |   32.7 |
> | VITO |    VideoNet   |  Swin-S |  81.3 |   46.1 | 49.8 |   33.5 |
> | Detcon-B |    ImageNet   |  Swin-S |  81.4  |    46.1  | 50.4 |   33.1 |
>
> 5. _“Current state-of-the-art for video recognition is achieved by initializing from pre-training on static images...”_
>
> We respectfully disagree that such experiments would be beneficial to this current study. By using an ImageNet pretrained backbone, it would be difficult to disentangle the effects of ImageNet pretraining from video pretraining. However, we appreciate the suggestion and believe that these experiments or co-training on image and video datasets would be a very fruitful future direction to pursue in future work.
>
> 6. _“Insufficient evaluations ... performance on classification ...  few-shot performance of such models?”_
>
> We have added an evaluation on DAVIS segmentation to demonstrate that VITO not only has strong scene understanding capabilities, but strongly surpasses ImageNet pretraining on video-based tasks (see general response Q1). _Note that this evaluation is 1-shot_: the model is provided with a single segmented frame and must extrapolate it to the rest of the sequence.
>
> In order to compare ImageNet- and video-pretraining fairly, we did not consider ImageNet classification to be a useful evaluation. ImageNet pretraining would help (somewhat trivially) in this case. The 4 tasks we choose instead are standard and high-quality benchmarks for gauging the effectiveness of image representations.

---

> > ### Author Response · Authors · 2022-11-17
> > **response (continued)**
> >
> > 7. _“How does this compare with imagenet-22k pretraining?...”_
> >
> > Thank you for this interesting suggestion, which would elucidate which aspects of our curation pipeline are important for our result. On the one hand, it may be that the distribution of video content need only coarsely match that of ImageNet, in which case filtering with a ImageNet-1k or 22k classifier would not change the results. However in expanding the vocabulary of the classifier we would increase the diversity of video content, which could improve the final representations. We will have to leave this very interesting question for future work.

---

### Official Review · Reviewer_kndf · 2022-10-23

**Confidence:** 5
**Correctness:** 4
**Technical Novelty And Significance:** 2
**Empirical Novelty And Significance:** 2
**Recommendation:** 3

**Clarity, Quality, Novelty And Reproducibility:**

The papers reads very well. I have my concerns about novelty and significance of experimental validation as explained above.

**Details Of Ethics Concerns:**

No ethics concerns

**Strength And Weaknesses:**

On the positive side:
+ The paper reports some results that might be useful for the community.
On the negative side:
- The methodological contributions are quite thin, mainly comprising minimal changes to existing strategies for augmentation and attention pooling (which is also very standard).
- The method is shown to work on the proposed curated dataset. Although the point of using a video dataset close to imagenet for a fair comparison is valid, in my opinion it also weakens the paper's contribution in a sense that learning from uncurated video data is a much more challenging and impactful goal.
- Experiments: from the experiments (table 1) it is not clear how VITO performs when trained on other video datasets (like Kinetics or Youtube 8M). Similarly, it is not obvious how other previously proposed methods would perform when trained on the newly proposed curated dataset. I suppose a strong baseline would be also to train DINO on the new dataset.

**Summary Of The Paper:**

This paper proposes a method to learn image-based representations from video. The paper makes two contributions: (1) on the network side, the authors introduce simple modifications to the way augmentations are applied (e.g. change the scale and uniform temporal sampling ) and a simple attention head for attention pooling to replace average pooling before the features are passed to the MLP. (2) on the data side, a new video dataset is proposed using a data curation process which tries to address the domain gap between imagenet and other video datasets. They evaluate on semantic segmentation and object detection after fine-tuning.

**Summary Of The Review:**

The paper reports some results that might be useful for the community, but I believe more work is required to increase the impact of the proposed method.

---

> ### Author Response · Authors · 2022-11-17
> **Specific response to reviewer kndf**
>
> We thank the reviewer for the constructive feedback and address their concerns point-by-point (reviewer comments in italics):
>
> 1. _”The methodological contributions are quite thin...”_.
>
> We agree that the components of VITO are standard on their own, but do believe that the idea of using multi-scale attention pooling for learning correspondences across frames is novel within this context. Specifically, it is a conceptual departure from recent efforts that have tried to do explicit object tracking to extract “views” for contrastive learning [1, 2] or methods that use explicit masks for a similar purpose in spatial contrastive learning [3].
>
> Furthermore, we consider this technical simplicity to be an asset of our approach given its empirical results. While the changes are simple, together they produce a result that is both novel and surprising: video pretraining closes the gap with ImageNet pretraining in scene understanding tasks, while strongly surpassing it in an evaluation of temporal understanding (DAVIS segmentation, see Q1 general response). In addition, the simplicity of our approach makes it easy to build upon and adapt with new architectures and self-supervised learning objectives. We hope this will open the door to more work on using videos for learning strong image representations.
>
> [1] Sharma et al. "Pixel-level Correspondence for Self-Supervised Learning from Video." ICML 2022
>
> [2] Xiong et al. "Self-Supervised Representation Learning from Flow Equivariance." ICCV 2021
>
> [3] Hénaff et al. "Efficient visual pretraining with contrastive detection." ICCV 2021
>
> 2. _”The method is shown to work on the proposed curated dataset... learning from uncurated video data is a much more challenging and impactful goal”_.
>
> While we understand the reviewer’s point of view, we respectfully disagree with the premise that learning from completely uncurated video data is in fact much more impactful or necessary, even though it may be more challenging. Rather we think our results suggest that more work should be focused on identifying ways to use data curation effectively in collecting large-scale datasets, perhaps in a totally unsupervised way.  Moreover, in contrast to ImageNet (which was manually curated) our current approach, which uses automatic curation, is scalable to arbitrarily large data collection.
> Nevertheless, we have added evaluations of our method on standard datasets: AudioSet, and the larger, uncurated YT8M dataset. In both cases, we strongly outperform recent prior work that train on these datasets:
>
> | **Method**      | **Dataset** | **PASCAL (Seg)**   | **ADE20K (Seg)**  | **COCO (Det)**   | **LVIS (Det)**  |
> | :---        |    :----:   |    :----: |    :----:   |    :----:  |   ---: |
> | MMV (Alayrac et al. 2020)   |    Audioset + HT   |    70.6 |   32.5 | 41.3 |  24.2 |
> | VITO |    Audioset   |    **73.6** |    **38.5** |  **43.2** |   **25.0** |
> | VIVI (Tschannen et al. 2020) |    YT8M   |    65.8 |    34.2 | 41.3 |   23.2 |
> | VITO |    YT8M   |    **71.8** |    **37.8** | **42.7** |   **24.6** |
>
> 3. _”from the experiments (table 1) it is not clear how VITO performs when trained on other video datasets ... not obvious how other previously proposed methods would perform ... train DINO on the new dataset”_.
>
> We appreciate the comment and hope to have answered this point in the response above by comparing VITO trained on Audioset and YT8M to recent top performing video pretraining methods that also train on these datasets. We believe this is a strong indication that it is not only our dataset, but also our method that contributes to our results.
> In terms of training competitive baselines on the VideoNet dataset, we also train MoCLR on VideoNet, and find that VITO strongly outperforms this baseline:
>
> | **Method**      | **Dataset** | **PASCAL (Seg)**   | **ADE20K (Seg)**  | **COCO (Det)**   | **LVIS (Det)**  |
> | :---        |    :----:   |    :----: |    :----:   |    :----:  |   ---: |
> | MoCLR |    VideoNet   |    73.5 |    38.0 | 43.1 |   24.8 |
> | VITO        |    VideoNet   |    **76.3** |   **39.4** | **44.0** |  **25.7** |
>
> While we do not train DINO on VideoNet, we believe MoCLR to be a strong enough baseline to be a good replacement for it. Indeed, when pretrained on ImageNet, MoCLR surpasses BYOL and DINO when transferring to PASCAL and ADE20K, and is close to DINO when transferring to COCO and LVIS (0.5% difference, see below). Therefore, it is safe to assume that VITO would also outperform DINO when pretrained on VideoNet.
>
> | **Method**      | **Dataset** | **PASCAL (Seg)**   | **ADE20K (Seg)**  | **COCO (Det)**   | **LVIS (Det)**  |
> | :---        |    :----:   |    :----: |    :----:   |    :----:  |   ---: |
> | Supervised |    ImageNet   |    71.3 |    33.5 | 44.2 |   25.2 |
> | BYOL        |    ImageNet   |    76.1 |   38.8 | 43.7 |  25.5 |
> | DINO |    ImageNet   |    76.1 |    39.0 | **44.3** |   **26.4** |
> | MoCLR        |    ImageNet   |    **76.4** |   **39.2** | 43.9 |  25.8 |

---

### Official Review · Reviewer_ACJo · 2022-10-24

**Confidence:** 4
**Correctness:** 3
**Technical Novelty And Significance:** 1
**Empirical Novelty And Significance:** 3
**Recommendation:** 5

**Clarity, Quality, Novelty And Reproducibility:**

Clarity: the paper is well written and easy to follow.

Originality: the paper is novel in an experimental sense (the first to match ImageNet pre-training using video frames), but there is no technical novelty.

Reproducibility: the authors are planning to release the code, but not the data (which is collected from the internet). Hence, reproducing their results will be very challenging, but, strictly speaking, not impossible.

**Strength And Weaknesses:**

Strengths:

The paper is well written and is easy to follow.

The proposed approach is sound and its effectiveness is convincingly confirmed by an ablation analysis.

The final model trained on curated video frames archives similar performance to models trained on ImageNet in terms of downstream semantic segmentation and object detection accuracy.


Weaknesses:

This is a purely empirical study, there is no technical contribution (all that techniques that are used have been introduced in the past).

Related work overview is incomplete. In particular, the authors ignore the whole line of work on space-time walks for self-supervised image representation learning from videos  which is highly relevant (e.g. Jabri et al., NIPS'21 but also prior works and follow ups).

In the experiments the authors do not control for the amount of training data. In particular, they have collected 1,180,042 videos each of which is at least 10 seconds long. How many frames is that? There is indeed a lot of redundancy between frames from the same video, but the redundancy is by far not 100%, especially in relatively long videos. For a fair comparison, the authors need to report results when only one pair of frames is sampled from each vides in a pre-processing step, so that the overall amount of data is comparable to ImageNet.

Most importantly, the value of using videos as a source of training data for image-level contrastive learning is not convincingly demonstrated. The authors have to go to great lengths to simply match the ImageNet performance. The downsides of this approach are clear (e.g. much higher storage requirements) but the benefits are not obvious from the paper.

**Summary Of The Paper:**

The authors adapt existing contrastive learning approaches for self-supervised image representation learning to learning from video frames instead of static images. In particular, they first adapt the data augmentation steps, which are key to contrastive learning, to the video domain. Two main adaptations are proposed: first, the crop size is increased to account for the fact that, unlike ImageNet, in-the-wild videos are not centered on a single object, hence very aggressive cropping can make the task unnecessary hard. Second, they utilize a multi-scale attention pooling module to account for the fact that the frame content can shift significantly over time. Finally, they collect a new dataset which is automatically curated (using an ImageNet classifier) to match the ImageNet category distribution. They then experimentally demonstrate that, by combining these model and data improvements, self-supervised pre-training on video frames can come close to pre-training on ImageNet in terms of downstream performance on the tasks of semantic segmentation and object detection.

**Summary Of The Review:**

This is a (for the most part) solid empirical study on adapting image-level contrastive learning methods to video frames. That said, a crucial factor is missing from the ablation analysis (control for the amount of training data) and the overall value of using video frames in this framework is not well justified. Moreover, related work overview has significant omissions. I hope the authors can address these concerns in the rebuttal.

---

> ### Author Response · Authors · 2022-11-17
> **Specific response to reviewer ACJo**
>
> We thank the reviewer for their detailed and useful comments and address their concerns point-by-point. (reviewer comments in italics).
>
> 1. _“This is a purely empirical study, there is no technical contribution ...”._
>
> We agree that some of the techniques we use have been introduced in the past in some form, but note that the idea of using multi-scale attention pooling for learning correspondences across frames is novel. Specifically, it is a conceptual departure from recent efforts that have tried to do explicit object tracking to extract “views” for contrastive learning [1, 2] or methods that use explicit masks for a similar purpose in spatial contrastive learning [3].
>
> Furthermore, we consider this technical simplicity to be an asset of our approach given its empirical results. While the changes are simple, they produce a result that is very novel: video pretraining closes the gap with ImageNet pretraining, while surpassing it in temporal understanding. It is also interesting that such simple a model strongly outperforms prior video pretraining methods that are far more complex.
>
> [1] Sharma et al. "Pixel-level Correspondence for Self-Supervised Learning from Video." ICML 2022
>
> [2] Xiong et al. "Self-Supervised Representation Learning from Flow Equivariance." _ICCV 2021_
>
> [3] Hénaff et al. "Efficient visual pretraining with contrastive detection." ICCV 2021
>
> 2. _“Related work overview is incomplete. In particular, the authors ignore the whole line of work on space-time walks for self-supervised image representation learning from videos  (e.g. Jabri et al. 2021)”._
>
> We thank the reviewer for highlighting the space-time random walks work and we will add it to the related work section and discuss its relation with our method. Briefly, this line of work focuses on learning a video graph representation that maximizes the likelihood of returning to the initial node (representation of one patch) when walking along a graph constructed from a 'palindrome' of patches from sequential frames. On the other hand, VITO discovers correspondences between pairs of frames using the contrastive loss and attention pooling, which we believe makes it simpler and more efficient to train as we do not need to ‘patchify’ frames or compute random walks across multiple frames. We do not quantitatively compare to Jabri et al. in our scene understanding evaluations because they did not evaluate on those benchmarks and only train ResNet-18 models (Xu & Wang [1] find that training ResNet-50 with Jabri et al’s method is unstable).
>
> Nevertheless, having run our additional evaluation on the DAVIS 2017 segmentation benchmark, we have confirmed that we outperform their method in that domain.
>
> | **Method**    | **J-Mean**   | **F-Mean**      |
> | :---        |    :----:   |    :----: |
> | VITO (ours)       |    **65.5** |    **70.8** |
> | Jabri et al. (VideoWalk)       |    64.6 |    70.6 |
>
> Additionally, in our paper we compare quantitatively (Table 1) to a conceptually related, more recent work, that also uses a cycle-consistency objective [2], and show that we outperform that method across all of our benchmarks.
>
> [1] Xu and Wang. Rethinking self-supervised correspondence learning: A video framelevel similarity perspective. _ICCV 2021_.
>
> [2] Wu and Wang. "Contrastive learning of image representations with cross-video cycle-consistency." _ICCV 2021_
>
> 3. _“In the experiments the authors do not control for the amount of training data...”._
>
> We respectfully point out that this is a common misconception: more data does not necessarily help self-supervised learning. We illustrate this (in our general response, Question 2) by running MoCLR on a dataset of 300M images, JFT, and find that it in fact performs significantly worse than ImageNet pretraining despite using 300x more data. Similarly, VITO strongly surpasses JFT pretraining even though it has access to a similar number of total frames.
>
> More important, therefore, is the distribution the data is sampled from, as indicated by the value of pretraining on VideoNet over AudioSet or YouTube-8M.
>
> 4. _“Most importantly, the value of using videos as a source of training data for image-level contrastive learning is not convincingly demonstrated...”_
>
> We agree with the reviewer and have added an evaluation to address this (see general response Q1). Specifically, we find that video pretraining with VITO yields better generalization to a video-based task (DAVIS segmentation) than a variety of ImageNet-pretrained models. Note that prior work on video pretraining had found similar benefits, however this has always come at the expense of performance on standard image benchmarks like scene segmentation and object detection. VITO is therefore the first video pretraining method that benefits from the best of both worlds: increased video understanding without sacrificing scene understanding. We hope this will encourage the community to use videos as the new default for learning visual representations.

---

> > ### Comment · Reviewer_ACJo · 2022-11-28
> > **Re:re**
> >
> > I thank the authors for their detailed response. It addresses some of my concerns, but only partially. In particular, I still find technical novelty to be marginal.
> >
> > Related work has been updated, and a comparison with Jabri et al. on DAVIS reported, but only in response to this review. In the corresponding Table 2 in the paper only weaker baselines are included (what a shame).
> >
> > The argument about the training set size in not convincing. Yes, the distribution of the data is critical, but this does not mean that the dataset size is not important. The authors are avoiding to report a simple fair comparison by sampling a single pair of frames from each video as a pre-processing step, which is also requested by another reviewer, further supporting my suspicion that the reported experimental comparisons are unfair.
> >
> > Overall, while the updated version of the manuscript makes a stronger argument in support of using videos to learn image-level representations, I still find it to be not convincing enough and intend to keep my score.

---

> > > ### Author Response · Authors · 2022-12-02
> > > **Response to Reviewer ACJo: training data size experiment**
> > >
> > > _I thank the authors for their detailed response. It addresses some of my concerns, but only partially. In particular, I still find technical novelty to be marginal._
> > >
> > > We accept that the reviewer feels the technical novelty is not enough but again emphasize that it is surprising that with our changes, we outperform all prior video pretraining on transfer to image tasks (even with comparable training datasets like R2V2). Importantly, we are competitive with ImageNet pretraining on image-tasks but also _outperform_ ImageNet pretraining on two video-based tasks (DAVIS segmentation and UCF-101). We feel that this clearly demonstrates the benefit of our method: the ability to learn a single image representation that can be effectively transferred to both image- and video-based tasks without separate architectures or training paradigms. We feel this empirical result is a great value-add to the community precisely because of the simplicity of the method compared to many of the prior works.
> > >
> > > _Related work has been updated, and a comparison with Jabri et al. on DAVIS reported, but only in response to this review. In the corresponding Table 2 in the paper only weaker baselines are included (what a shame)._
> > >
> > > Thank you for pointing out that we missed this inclusion in the paper revision. We will add the comparison to Jabri et al to the final version of the paper.
> > >
> > > _The argument about the training set size in not convincing. Yes, the distribution of the data is critical, but this does not mean that the dataset size is not important. The authors are avoiding to report a simple fair comparison by sampling a single pair of frames from each video as a pre-processing step, which is also requested by another reviewer, further supporting my suspicion that the reported experimental comparisons are unfair._
> > >
> > > We take issue with the idea that we are purposefully making unfair comparisons. We provided the intuition about data size vs data distribution, since it is clear that the data distribution matters far more, as evidenced by the fact that a 300M JFT dataset performs worse than ImageNet for pretraining. Similarly, pretraining on YouTube8M (which is much bigger than VideoNet) performs significantly worse than VideoNet pretraining.
> > >
> > > Nevertheless, we followed the reviewer’s suggestion and ran VITO on a version of VideoNet with a single pair of frames per video (using the “delta” sampling scheme for simplicity). In the table below, we present our best “marginal sampling” method, the “delta sampling” method where each video uses a different pair of frames but with a fixed time delay, and finally the “delta sampling single pair” where we pre-filter the dataset to choose a single pair of frames from each video with a fixed time-difference.  The results are as you would expect: performance degrades slightly, but not significantly on all benchmarks and all our main conclusions remain unchanged. Even with the “delta sampling single pair” scheme, VITO outperforms all prior video pretraining by a large margin and is competitive with ImageNet pretraining. This confirms the fact that VITO attains its performance thanks to_ the distribution of VideoNet videos, and its learning paradigm_, not simply a larger number of frames. We hope this alleviates the concerns regarding the issue of data size, and we will include this ablation in the final version of the paper.
> > >
> > > | **Method**      | **Dataset** | **PASCAL (Seg)**   | **ADE20K (Seg)**  | **COCO (Det)**   | **LVIS (Det)**  |
> > > | :---        |    :----:   |    :----: |    :----:   |    :----:  |   ---: |
> > > | VITO (best)       |    VideoNet, full   |    76.3 |   39.4 | 44.0 |  25.7 |
> > > | VITO (delta sampling) | VideoNet, full | 75.8 | 39.4 | 43.8 | 25.7 |
> > > | VITO (delta sampling single pair) | VideoNet, single pair of frames per video | 75.2 | 39.3 | 43.5 | 25.9 |
> > > | MoCLR |    ImageNet   |    76.4 |    39.2 | 43.9 |   25.8 |

---

> > > > ### Comment · Reviewer_ACJo · 2022-12-02
> > > > **Re:re**
> > > >
> > > > I thank the authors for providing the missing ablation. The results demonstrate that in a fair comparison (the last two rows in the table) pertaining on videos indeed does worse than pre-training on images when transferring to the most common image-level benchmarks (PASCAL and COCO). To achieve the same level of performance orders of magnitude more video frames are needed compared to static images.
> > > >
> > > > I do see value in the proposed approach as it allows to learn representations that achieve top performance on downstream video tasks while maintaining comparable, even if somewhat lower, performance on image-level benchmarks. That said, using image-level representations for video understanding is not a promising direction to begin with, so the motivation of this paper remains questionable in my opinion. Overall, I don't have a strong recommendation for this paper: it meets the conference standards, but I'm ok with rejecting it as well.

---

> > > > > ### Author Response · Authors · 2022-12-09
> > > > > **further response to ACJo**
> > > > >
> > > > > We thank the reviewer for the quick response and taking the time to go through the results. We did want to briefly address one point regarding the results of this last ablation.
> > > > >
> > > > > _The results demonstrate that in a fair comparison (the last two rows in the table) pertaining on videos indeed does worse than pre-training on images when transferring to the most common image-level benchmarks (PASCAL and COCO)._
> > > > >
> > > > > We do accept there is a drop in performance when transferring to PASCAL and COCO, however, we note that on the more recent and challenging benchmarks (ADE20K and LVIS) we in fact slightly _outperform_ ImageNet pretraining. ADE20K and LVIS are now well-accepted benchmarks, proposed by the community to address the shortcomings of PASCAL and COCO respectively. Therefore we don't think the results on those two datasets should be undervalued in any way compared to PASCAL and COCO. If anything, ADE20K and LVIS have become the more standard benchmarks for visual understanding: the COCO Challenge has been deprecated in favor of LVIS since 2021, and all recent representation learning methods (e.g. masked auto-encoding and variants) evaluate on ADE20K but not PASCAL.
> > > > >
> > > > > _That said, using image-level representations for video understanding is not a promising direction to begin with,_
> > > > >
> > > > > We would like to remind the reviewer that the focus of our work was learning general image representations that can perform well at a variety of image- and video-based tasks. This approach is specifically advantageous in that it does not require multiple separate models trained on image and video datasets independently. Nevertheless, specifically for video understanding, we believe our approach could seamlessly be extended to utilize standard video architectures (processing video clips instead of single frames), which we consider to be a fruitful direction for future work.

---

### Official Review · Reviewer_iRvv · 2022-10-25

**Confidence:** 4
**Clarity, Quality, Novelty And Reproducibility:** 1. it's not clear what `we sample fro…
**Correctness:** 3
**Technical Novelty And Significance:** 2
**Empirical Novelty And Significance:** 2
**Recommendation:** 6

**Strength And Weaknesses:**

Pros:

This paper is well-organized and easy to follow. The technical contribution and the way to do data curation make sense to me. The ablation is comprehensive and the final results compared with other video-based model looks good, which shows the effectiveness of the proposed framework.

Cons:

The motivation of this paper is not very clear, as the results using video pretraining do not show superior performance compared with image pretraining based on the benchmark the authors chose. If the authors believe video is beneficial maybe choose another benchmark (like video segmentation or tracking) or at least combine ImageNet with the proposed VideoNet data to showcase better performance compared with the ImageNet baseline.
For other misc comments see the section below.


**Summary Of The Paper:**

This paper presents a video self-supervised pretraining pipeline called VITO. It made several modifications over existing contrastive learning frameworks including larger crop size, improved temporal sampling scheme, and multi-scale attention feature pooling for the projector. The authors also investigated the data domain mismatch and propose a video curation procedure to further improve the model pretraining performance. Experimental results on semantic segmentation and object detection show that the proposed framework can achieve better results among methods pretrained on video datasets, and be competitive with the models pretrained on ImageNet.

**Summary Of The Review:**

Overall I believe this paper makes novel technical contributions (using learnable attention gates to align features from different frames, pointing out that the video data quality actually matters a lot for existing models on several benchmarks, etc.). However, the evaluation part is my major concern, would be great if the authors could better motivate why video is important (although this might be shown in other works) and show either video+image could perform better than image only, or video pretraining learn better representations on some tasks compared with image-based and other video-based baselines.

---

> ### Author Response · Authors · 2022-11-17
> **Specific response to reviewer iRvv**
>
> We thank the reviewer for their detailed and useful comments and will now address the stated concerns (reviewer comments in italics).
>
> _"The motivation of this paper is not very clear, as the results using video pretraining do not show superior performance compared with image pretraining based on the benchmark the authors chose. If the authors believe video is beneficial maybe choose another benchmark (like video segmentation or tracking)"_
>
> We agree with the reviewer and have added an evaluation showing that video pretraining with VITO yields better generalization to a video-based task than ImageNet pretraining (DAVIS video segmentation, see Question 1 above). This shows that, with VITO, video pretraining yields strictly more general representations than image pretraining, in that they benefit from both spatial and temporal understanding. Image pretraining lacks temporal understanding, and prior work on video pretraining has failed to recover spatial understanding competitive with image pretraining.
>
> We will add these evaluations and use them to clarify our motivation in the final version of the text.
>
>
>
> Now to address the more specific concerns:
>
> 1.  _"it's not clear what we sample from a uniform distribution over the entire video segment of length T = 2.56s means,”_
>
> We start by sampling a video segment consisting of 64 frames (which lasts 2.56s when played at 25 FPS), and randomly sample two frames from this segment (using a uniform distribution). We will clarify this in the text.
>
>  _"It's also not clear to me why it is good that the mode of the distribution is at 0 (according to Figure A.1), wouldn't it be a better choice to make the distribution of time delays uniform?"_
>
> We believe that having the most frequent time difference be 0 is sensible because longer time differences increase the likelihood of a change in scene content, and thus a false positive. Note that this is analogous to other augmentations used in SSL, e.g. random translations and color jittering are all centered at zero. Beyond this intuition, we empirically validate that this sampling scheme is more effective than other sampling schemes (Figure A.1). A uniform distribution of time delays underperforms in the multi-view setup, we will add this explicit comparison.
>
> 2) _“Using an ImageNet-pretrained model to do data curation is less preferable from my perspective. Although I agree that it might be necessary. this inevitably obscures the boundary between semi-supervised and self-supervised learning. ”_
>
> We appreciate this point, but do not see how our method is qualitatively different from what is well accepted for evaluating self-supervised algorithms in Imagenet, which was manually curated. Instead of manual data curation, we are using an automated approximation that makes our approach scalable to arbitrarily large data collection.
> We note that this data curation is equally important for image pretraining, and find models pretrained on much larger, but uncurated datasets such as JFT-300M to perform substantially worse (see Question 2 above).
>
> 3) _Xiong, Yuwen, et al. "Self-supervised representation learning from flow equivariance." Proceedings of the IEEE/CVF International Conference on Computer Vision. 2021._
>
> Thank you for the reference, we will add it to the related work section and discuss its relation with our method. Briefly, whereas this work infers positive pairs across time using a pretrained optical flow network, VITO discovers correspondences between frames using the contrastive loss and attention pooling, which we believe makes it simpler, more efficient, and possibly more generalizable to new datasets.
> Beyond this, a side-by-side comparison of both methods is difficult as the authors primarily evaluate their method in autonomous driving scenarios and do not provide pretrained checkpoints.

---

> > ### Comment · Reviewer_iRvv · 2022-12-05
> > **Reply to authors**
> >
> > I appreciate the authors' response and also checked other reviewers' comments. Overall I agree with Reviewer ACJo's latest comments: this paper shows good empirical results, but the motivation for the dataset part is not very solid as I do not think making an ImageNet-like video dataset is very valuable and exciting. Thus I'll keep my rating as marginal accept.

---

> > > ### Author Response · Authors · 2022-12-09
> > > **response to reviewer iRvv**
> > >
> > > We thank the reviewer for reading our updated results and comments.
> > >
> > > _the motivation for the dataset part is not very solid as I do not think making an ImageNet-like video dataset is very valuable and exciting._
> > >
> > > While we appreciate the reviewer's opinion, we would again like to reiterate that the value of the dataset is just one piece of the work, and we have demonstrated that our methodology in fact still outperforms all prior video pre-training work on all downstream evaluations even when trained on standard datasets.
> > >
> > > Nevertheless we do feel that our experiments demonstrating the impact of the image content distribution vs. amount of total training data is a valuable contribution to the field. It has long been thought that simply scaling self-supervised contrastive methods to larger datasets would immediately yield better generalization and transfer performance. However, we show that scaling dataset size without caring about the image distribution in fact leads to marginal (or sometime no) benefits. Therefore, we believe that constructing methods to curate large-scale video (and image) datasets efficiently, without manual annotation, will become increasingly more important.

---

### Author Response · Authors · 2022-11-17
**General response to all reviewers**

We thank all of the reviewers for their insightful and constructive comments. This comment addresses a few common themes in the reviews.

**Question 1: Why learn image representations from videos? Why not just train on ImageNet?**

We have added an analysis showing that VITO representations are more general than ones trained on ImageNet: _in addition to being useful for scene understanding tasks, these representations also are much more effective in video-based tasks_. We measured this generalization by segmenting DAVIS videos with features learned with VITO on VideoNet, or a variety of different ImageNet pretraining methods. VITO surpasses all of them:

**DAVIS 2017**
| **Method**      | **Dataset** | **J-Mean**   | **F-Mean**      |
| :---        |    :----:   |    :----: |    ---: |
| DetCon       |    ImageNet   |    63.1 |    66.4 |
| MoCLR        |    ImageNet   |    63.1 |    67.8 |
| BYOL        |    ImageNet   |    63.8 |    69.4 |
| VITO        |    VideoNet   |    **65.5** |    **70.8** |

Note that previous works had also found that video pretraining yields better performance on video-based tasks than image pretraining, yet they did so by sacrificing significant performance on canonical image understanding benchmarks. Using video pretraining, VITO, for the first time, achieves expected performance gains on video-based tasks, while maintaining ImageNet-level performance on scene understanding (segmentation and detection) (paper Table 1). We will adapt the text to reflect this.

**Question 2: does video pretraining help just because there are more frames in a video?**

Reviewers brought up the concern that even though our VideoNet dataset is matched in the number of training examples to ImageNet, the fact that it contains multiple frames per clip could make the comparison unfair. Indeed, the VideoNet dataset had ~1.2 million video clips which are 10s long and sampled at 25 FPS, yielding 300M frames in total.

To control for the number of frames, we additionally trained our MoCLR image pretraining baseline on the JFT dataset, which also contains 300 million frames. Despite the fact that many of the VideoNet frames are highly redundant, VITO outperforms the JFT pretrained model, indicating that total numbers of frames is not a driving factor of our results.

| **Method**      | **Dataset** | **PASCAL (Seg)**   | **ADE20K (Seg)**  | **COCO (Det)**   | **LVIS (Det)**  |
| :---        |    :----:   |    :----: |    :----:   |    :----:  |   ---: |
| VITO        |    VideoNet   |    76.3 |   39.4 | 44.0 |  25.7 |
| MoCLR |    ImageNet   |    76.4 |    39.2 | 43.9 |   25.8 |
| MoCLR |    JFT   |    74.3 |    38.7| 43.2 |   25.4 |

**Question 3: what is the value of our data curation methodology and does this limit the impact of our results?**

In the creation of the VideoNet dataset, we use an automated method to curate the data distribution (applying an ImageNet classifier to frames to make sure the stated objects appear in the selected videos). Reviewers were concerned that this limits the impact of our method since we do not learn from totally uncurated datasets. While we appreciate this point, our method is not qualitatively different from what is well accepted for evaluating self-supervised algorithms on images (i.e. ImageNet). Instead of manual data curation, we use an automated approximation that makes our approach scalable to arbitrarily large data collection.

We believe that the VideoNet results are impactful for the community,  but we have added evaluations that demonstrate that our methodology still outperforms prior video pretraining work that use existing video datasets (AudioSet and YT8M):

| **Method**      | **Dataset** | **PASCAL (Seg)**   | **ADE20K (Seg)**  | **COCO (Det)**   | **LVIS (Det)**  |
| :---        |    :----:   |    :----: |    :----:   |    :----:  |   ---: |
| MMV (Alayrac et al. 2020)   |    Audioset + HT   |    70.6 |   32.5 | 41.3 |  24.2 |
| VITO |    Audioset   |    **73.6** |    **38.5** |  **43.2** |   **25.0** |
| VIVI (Tschannen et al. 2020) |    YT8M   |    65.8 |    34.2 | 41.3 |   23.2 |
| VITO |    YT8M   |    **71.8** |    **37.8** | **42.7** |   **24.6** |

We note that data curation is equally important to obtain SoTA self-supervised image pretraining, and find models pretrained on much larger, but uncurated datasets like JFT perform substantially worse (see Question 2 and Tian et al. 2021).

Finally, we believe that there is a purely unsupervised curation approach that may be possible in which we simply measure the similarity of our dataset and ImageNet images (without labels) in an unsupervised embedding space to better align the two, rather than using a classifier. We leave this for future work, but hope that our results suggest that there is a potentially large benefit in developing unsupervised forms of data curation rather than trying to only learn from totally uncurated datasets.

---

> ### Author Response · Authors · 2022-11-17
> **Emphasizing comparison to prior work**
>
> For ease of access for the reviewers, we provide a section of our original paper results, comparing to prior published work in the space of video pretraining for learning image representations. Given the many comparisons in our Table 1, we wanted to highlight specifically the gap between VITO and all prior video pretraining work as this may have been understated in the text:
>
> | **Method**      | **Dataset** | **PASCAL (Seg)**   | **ADE20K (Seg)**  | **COCO (Det)**   | **LVIS (Det)**  |
> | :---        |    :----:   |    :----: |    :----:   |    :----:  |   ---: |
> | VFS       |    K400   |    63.9 |    31.4 | 41.6 |    23.2 |
> | VIVI        |    YT8M   |    65.8 |    34.2 |41.3 |    23.2 |
> | VINCE        |    R2V2   |    69.0 |    35.7 |42.4 |    24.4 |
> | CycleContrast |    R2V2   |    69.2 |    35.6  |42.8 |    24.5 |
> | MMV        |    Audioset + HowTo   |    70.6 |  32.5  |41.3 |    24.2 |
> | VITO        |    VideoNet   |    **76.3** |   **39.4** | **44.0** |  **25.7** |
> | MoCLR |    ImageNet   |    **76.4** |    **39.2** | **43.9** |   **25.8** |

---

### Author Response · Authors · 2022-11-18
**Final paper revision notes**

We thank the reviewers again for all of their constructive feedback and have tried to incorporate as many of the suggestions as possible in the revised paper submission. To summarize the new experiments we have added:
1. As noted below we have shown that our method outperforms strong ImageNet pretraining baselines on Davis 2017 Segmentation (Table 2).
2.  **In addition, we have also added a new evaluation on the UCF-101 action recognition task** (Table 3). We show that VITO outperforms prior 2-d (image) architectures by a large margin and even outperforms some recent specialized video architectures (3-d convolutions and variants).
3. We have added to the paper various other requested experiments regarding evaluating VITO on other uncurated video datasets, and scaling to larger model architectures (Swin transformers).

We also hope to have revised the text to emphasize the empirical novelty of the work and address concerns over the motivation and citation of additional related work.

---

### Decision · Program_Chairs · 2023-01-20

**Decision:**

Reject

**Justification For Why Not Higher Score:**

1. Limited technical contribution
2. Unconvincing experiments

**Justification For Why Not Lower Score:**

NA

**Metareview: Summary, Strengths And Weaknesses:**

This paper was reviewed by four experts in the field and received a mixed score. The main concerns are the limited novelty, unconvincing experiments, and lack of clarity. The authors did a good job of rebuttal and addressed many of the concerns. However, the reviewers (including all positive ones) still feel that more work is needed to get it to the best version. AC also agrees that this work can be much stronger with additional experiments. While this paper clearly has merit, the decision is not to recommend acceptance. The authors are encouraged to consider the reviewers' comments when revising the paper for submission elsewhere.